| **Open Peer Review** | Immunology | Letter to the Editor

# Vaccination caused an underestimation of the cumulative incidence of SARS-CoV-2 infection

Nana Owusu-Boaitey[1]

**KEYWORDS**  SARS-CoV-2, serology, vaccines

I read with interest the manuscript by Kao et al. (1). The authors used anti-nucleocapsid (anti-N) seropositivity to detect prior infection by severe acute respiratory syndrome coronavirus 2 (SARS-CoV-2). Lower anti-N seroprevalence occurred in a nationwide blood donor study compared to a nationwide commercial laboratory study (NBDS and NCLS, respectively). The authors aptly noted that antibody waning did not explain this difference (1); the direct assays used maintain high sensitivity with time following infection, in the absence of SARS-CoV-2 vaccination (2).

The authors provided evidence of a higher SARS-CoV-2 vaccination rate in the sampled NBDS population versus the NCLS population. I suggest this anti-spike (anti-S) vaccination likely reduced anti-N assay sensitivity in the NBDS sample, consistent with prior studies (3–6). Unfortunately, anti-N seropositivity was assessed in NBDS starting in September 2021 after vaccination campaigns began (7). It remains unclear whether differences in anti-N seroprevalence for NBDS versus NCLS existed prior to the onset of widespread vaccination.

Other research, however, confirmed that vaccinated blood donors were less likely than unvaccinated blood donors to be anti-N seropositive after infection (6). Prioritization of vaccination for the elderly would similarly decrease anti-N sensitivity in this age group. This explanation contradicts commentary that used NBDS anti-N seroprevalence data to argue for less "hybrid immunity" in the elderly compared to younger age groups (8). Higher anti-N seroprevalence in younger versus older age groups may therefore reflect age-specific differences in vaccination rates instead of differences in susceptibility to infection (9). There should thus be caution when comparing anti-N-based estimates for populations that differ in vaccination rates (10).

Blood donor studies such as NBDS highlight the importance of vaccine-induced decreases in anti-N assay sensitivity. This sensitivity decrease impacts interpretation of serological analyses in other contexts as well. Vaccine-induced decreases in anti-N sensitivity, for example, impaired detection of breakthrough infections in vaccine trials (3). Booster vaccination further decreased sensitivity, though this effect was attenuated with time following administration of a booster dose (6, 11). Vaccination also caused anti-N serology to underestimate the cumulative incidence of SARS-CoV-2 infection during testing of the general population (5). Reduced anti-N assay sensitivity would further inflate severity estimates inferred from cumulative incidence, such as fatality rate from infection (IFR) and hospitalization rate from infection (IHR). Adjustment for anti-N sensitivity is thus needed for serology-based assessment of IFR and IHR changes with time following introduction of vaccination (10).

Prompt neutralization of SARS-CoV-2 by vaccine-induced anti-S antibodies may forestall the generation of anti-N antibodies (4, 6), or anti-S vaccination may select for B cells that target spike protein epitopes instead of nucleocapsid protein epitopes.

Address correspondence to Nana Owusu-Boaitey, nana3@umd.edu.

The authors declare no conflict of interest.

See the original article at https://doi.org/10.1128/spectrum.00123-24.

Care should therefore be taken when interpreting infection-induced seroprevalence for antibodies targeting antigens different from those selected as vaccination targets.

## ACKNOWLEDGMENTS

This work was not supported by any specific grant from funding agencies in the public, commercial, or not-for-profit sectors.

## AUTHOR AFFILIATION

[1]Fischell Department of Bioengineering, University of Maryland, College Park, Maryland, USA

## AUTHOR ORCIDs

Nana Owusu-Boaitey  http://orcid.org/0000-0003-0974-6346

## ADDITIONAL FILES

The following material is available online.

Open Peer Review

**PEER REVIEW HISTORY (review-history.pdf).** An accounting of the reviewer comments and feedback.

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
