## [Reviewer comments · Microbiology Spectrum]

Microbiology Spectrum

Vaccination caused underestimation of cumulative incidence of SARS-CoV-2 infection

Nana Owusu-Boaitey

Corresponding Author(s): Nana Owusu-Boaitey, University of Maryland

Review Timeline:

Submission Date:	July 4, 2024
Editorial Decision:	September 10, 2024
Revision Received:	September 10, 2024
Accepted:	September 11, 2024

Editor: Gabriel Parra

Reviewer(s): The reviewers have opted to remain anonymous.

Transaction Report:

DOI: <https://doi.org/10.1128/spectrum.01558-24>

Re: Spectrum01558-24 (Vaccination caused underestimation of cumulative incidence of SARS-CoV-2 infection)

Dear Dr. Nana Owusu-Boaitey:

Thank you for the privilege of reviewing your work. Below you will find my comments, instructions from the Spectrum editorial office, and the reviewer comments.

I am pleased to inform you that your manuscript has been editorially accepted for publication. However, there are a few additional questions in the submission form that need to be answered before the final decision. Once these are completed, please return your submission so that I can move your paper forward to acceptance.

Revision Guidelines

Sincerely,
Gabriel Parra
Editor
Microbiology Spectrum

Reviewer #2 (Comments for the Author):

The main point being made is that vaccination may blunt the anti-N response to infection, hence infections may be missed with anti-N testing in vaccinated individuals. This is an interesting and valid perspective.

1 Reviewer #2 (Comments for the Author):

2

3 The main point being made is that vaccination may blunt the anti-N response to infection, hence
4 infections may be missed with anti-N testing in vaccinated individuals. This is an interesting and
5 valid perspective.

6

7 Response: I thank the reviewer for this comment.

8

Re: Spectrum01558-24R1 (Vaccination caused underestimation of cumulative incidence of SARS-CoV-2 infection)

Dear Dr. Nana Owusu-Boaitey:

Thank you for submitting your letter to Microbiology Spectrum and responses to reviewer's comments. Your letter is now accepted for publication and will be published together with a reply from the authors of the original paper (Kao et al. Microbiol Spectr. 2024 Aug 6;12(8):e0012324).

Your paper will first be checked to make sure all elements meet the technical requirements. ASM staff will contact you if anything needs to be revised before copyediting and production can begin. Otherwise, you will be notified when your proofs are ready to be viewed.

Sincerely,
Gabriel Parra
Editor
Microbiology Spectrum